# Fundamental Research on Detecting Contradictions in Requirements: Taxonomy and Semi-Automated Approach

**Alexander Elenga Gärtner [1,2,\*], Tu-Anh Fay [2]**  **and Dietmar Göhlich [2]**

1   IAV GmbH, 10587 Berlin, Germany
2   Methods of Product Development and Mechatronics, Technische Universität Berlin, 10623 Berlin, Germany; tu-anh.fay@tu-berlin.de (T.-A.F.); dietmar.goehlich@tu-berlin.de (D.G.)
\*   Correspondence: alexander.elenga.gaertner@campus.tu-berlin.de or alexander.elenga.gaertner@iav.de

**Abstract:** Requirements documents can contain several thousand individual requirements. They must be error-free to avoid unnecessary complications and costs in the later product development stages. An important part of this is to identify contradictions between two requirements. The first step is therefore to define what contradictions are and in what form they can occur in requirement documents. In this paper the scientific theories regarding contradictions are discussed, concerning to their usefulness for the topic. In doing so, the Aristotelian Logic proved to provide the best basis for an application in the Requirements Engineering context. Based on this theory, we have created specific subtypes of contradictions to match them to the requirements engineering field. The identification of these subtypes is done by a formalization of the requirement sentences and a subsequent analysis by means of simple questions. To validate the method, industrial requirement documents were searched for contradictions. For each detected type of contradiction, we present an example of the detection process. Thereby, we show that the method is easy to apply and may also be used by non-specialists. Thus, our method provides a taxonomy as a basis for further research on automated contradiction detection as well as on automated quality analysis of requirements documents.

**Keywords:** requirements engineering; contradictions; conflicts; logic

## 1. Introduction

Complete and error-free requirements specifications are crucial for effective product development. One aspect of this, is to ensure that the documents are contradiction-free. On the way toward this, contradictions must first be defined in the Requirements Engineering (RE) context to recognize and classify them. Subsequently, the quality of the requirements specification can be determined and, depending on the class of the contradiction, a solution can be pursued.

### 1.1. Problem

Requirements form the basis for project planning, risk management, acceptance testing, and many other fields [1]. Requirements specifications that describe an entire system are often written in an interdisciplinary manner. The partial results must merged according to their logical and temporal dependencies, to form the overall solution. This process involves a risk of error, especially for complex systems, regarding the consistency of the partial solutions within the overall solution [2]. Therefore, it is not surprising that errors, e.g., in the form of contradictions, are often found in these documents.

Empirical research on requirements quality focuses on improvement techniques, with very few primary studies addressing evidence-based definitions and evaluations of quality attributes [3].

### 1.2. Contribution

In this paper, a proposal is made on what contradictions are in the RE context, how they can be classified, and how they can be determined. The classification of distinctive categories allows for a more consistent assessment of the quality of requirements documents, as different types of contradictions have different criticality levels. Also, depending on the type of contradiction, different approaches are needed to solve them. Finally, the proposed standardized solution provides a partially automated method, which is the basis for a potential fully automated contradiction detection.

Within our validation, examples from real requirements documents are used.

## 2. Fundamentals

The market study from Luisa et al. concluded that in most cases (95%) requirements documents were expressed in Natural Language [4], which is inherently ambiguous [5] and must therefore be interpreted to a certain degree. This interpretation can often be an undetected source of errors.

In this paper, we focus on contradictions between pairs of text-based requirements in tabular form. Below, we will specify how these requirements should be phrased and how contradictions are generally defined.

### 2.1. Formulation and Building Blocks

Ideally, requirements should be based on a specific scheme [1]: Requirement Expression = Boilerplate + Placeholder values. A boilerplate for a typical non-causal requirement shows the following form: The *<stakeholder type>* must be able to *<capability>*. Another example for a boilerplate could look like this: If *<operational condition (cause)>*, the *<system>* shall *<function>* not less than *<quantity> <object>*, e.g. If *the fuel tank is empty, the Flexray* shall su*stain communication* not less than *1 h*. Simplified, sentences are built with *<Cause>* + *<Effect>* which in turn consist of variables and conditions, as shown in Figure 1:

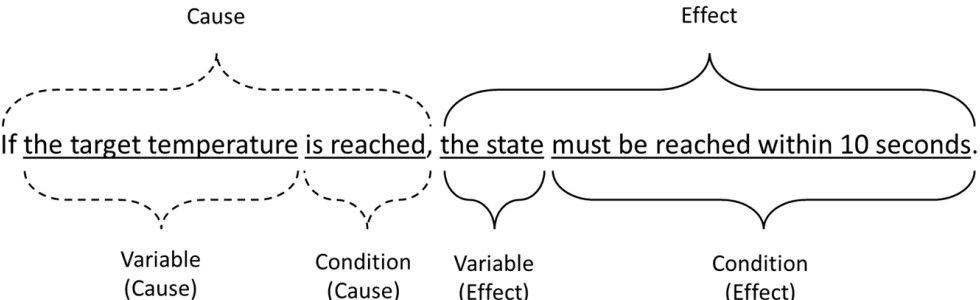

**Figure 1.** Formulation and building blocks.

### 2.2. Contradictions

In this paper, we differentiate between the term contradiction—which occurs when a statement is in opposition either with itself or an established fact—and the term contradictory, which we will explain below. In the literature on Requirements Engineering, many definitions of contradictions can be found, see Section 3 *Related Work.* To get the most generically valid and scientifically accepted definition, we base our theory on the logical philosophy of Aristotle. The foundation of his logic—also known as term logic, traditional logic, or formal logic—developed in his work Metaphysics is the law of non-contradiction (LNC) [6]. There, he argues that it is impossible that the same thing belongs and does not belong simultaneously in an identical way to the same object [7]. "The doctrine of the square of opposition [as seen in Figure 2; note by the author] originated with Aristotle in the fourth century BC and has occurred in logic texts ever since. Although severely criticized in recent decades, it is still regularly referred to" [8] and will hence serve as a basis for our purposes.

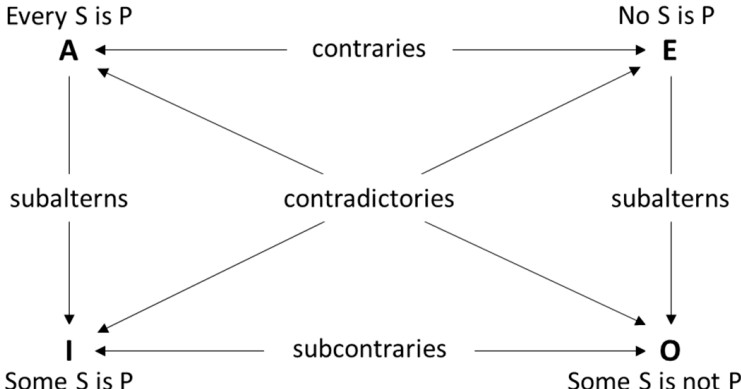

**Figure 2.** Square of opposition.

The relations can be described as follows:

- "Every S is P" and "Some S is not P" are contradictories.
- "No S is P" and "Some S is P" are contradictories.
- "Every S is P" and "No S is P" are contraries.
- "Some S is P" and "Some S is not P" are subcontraries.
- "Some S is P" is a subaltern of "Every S is P".
- "Some S is not P" is a subaltern of "No S is P".

Therefore, we have four main oppositions: contradictories, contraries, subcontraries, and subalterns.

1. Contradictory opposites, e.g., "he is sick"/"he is not sick", are mutually exhaustive and mutually inconsistent. This means, that one statement must be true and the other false or vice versa. They cannot both be true or false at the same time.
2. Contrary opposites, e.g., "it is black"/"it is white", are also mutually inconsistent, but not exhaustive. While they cannot both be true, they can both be false.
3. Subcontraries, e.g., "you can—If you want to—call in sick"/you can—If you want to—not call in sick" are mutually consistent. While they can simultaneously be true at the same time, they cannot simultaneously be false at the same time.
4. The statement "some people are sick" is the subaltern of "everybody is sick", while the latter is the superaltern of the former. If the superaltern is true, the subaltern must also be true and if the subaltern is false, the superaltern must also be false.

By these definitions, the four central kinds of opposition—contradictory, contrariety, subcontrariety, and subaltern—are mutually inconsistent.

In addition to the contradictions considered so far, there are other types. Kesselring differentiates between the Aristotelian LNC-contradictions, dialectic contradictions, and antinomies [9].

Dialectic contradictions are comparable to antagonisms or so-called „conflict of goals". For example:

- The vehicle must have high performance
- The vehicle must have low consumption

They don't stand in a mathematical/logical conflict but are incompatible in practice.

Antinomies denote conceptual or propositional structures in which the truth value oscillates. A famous example is: Plato says, "Socrates speaks the truth," and Socrates says, "Plato lies." They are often confused with self-contradictions [9].

In this paper, we will tackle the LNC conflicts, except for the subcontraries. As they can be valid simultaneously, subcontraries are not contradictions that need to be resolved for the RE work.

## 3. Related Work

In this section, we assembled related works in terms of classification of conflicts, detection of antagonisms, natural language processing for detecting conflicts, and finally ontology-based approach for detecting conflicts. In general, we saw a lack of real validation in these topics, as it is difficult to find non-academic institutions that are willing to share their requirements-documents for scientific analyses [10].

### 3.1. Classification of Conflicts

A classification of conflicts is suggested by Marneffe et al., in antonymy, negation, or numeric mismatches [11]. Negations and numeric mismatches do not fit the LNC classification of Aristotle and can only be partially combined.

Lamsweerde et al. are classifying conflicts into nine categories [12]:

1. Process-Level Deviation: Conflict between a process-level rule and a specific process state.
2. Instance-Level Deviation: Inconsistency between a product-level requirement and a specific state of the running system.
3. Terminology Clash: Usage of different terms for the same event
4. Designation Clash: Usage of the same term for different events
5. Structure Clash: Different explanations for a single real-world concept
6. Conflict: Two assertations are directly logically inconsistent
7. Divergence: Two assertations are indirectly (through a boundary condition) logically inconsistent
8. Competition: Particular case of the divergence
9. Obstruction: Another particular case of the divergence

This doesn't represent a classification with a consistent structure, since on the one hand the system level is taken as classification criteria and on the other hand the context is taken as classification criteria. Also, 7, 8, and 9 cannot clearly be differentiated.

Marneffe et al. propose a looser classification than ours. "Pairs such as 'Sally sold a boat to John' and 'John sold a boat to Sally' are tagged as contradictory" [11]. In the context of requirements engineering though, this should not be interpreted as a contradiction. This becomes clear in the following example: "Control unit 1 sends a signal to control unit 2. Control unit 2 sends a signal to control unit 1." It becomes more complicated when it says, "Control unit 1 sends the signal X to control unit 2. The control unit 2 sends the signal X to the control unit 1." This would indeed be a contradiction, but it would be classified as a dialectic contradiction: theoretically, it is possible, but it wouldn't make any practical sense.

Guo et al. propose a classification into three basic conflict types—inconsistencies, inclusions, and interlocks—which in turn can be divided into seven subcategories [13]. Inconsistencies are defined as contradictions between requirements that cannot both be fulfilled at the same time. Compared to LNC, this could correspond to contradictories or contraries as well as dialectical contradictions. Inclusions can correspond to both contradictories and contraries. Interlocks can be compared with subalterns. This represents a promising approach and to a large extent can be combined with LNC. In Section 4.2 *Contradictions—Subcategories*, parts of this classification are taken up, placed in the logical context, and further refined.

### 3.2. Natural Language Processing for Detecting Conflicts

According to Zhao et al., most of the studies (67.08%) in the Nature Language Processing domain combined with RE are "solution proposals, assessed by a laboratory experiment or an example application, while only a small percentage (7%) are assessed in industrial settings" [14], so they rarely have a practical validation.

However, to the best of our knowledge, no Machine-Learning oriented studies tried to classify or find contradictions. Many papers deal with classifying requirements, for example in functional and non-functional requirements [15] or in security-related requirements [16].

### 3.3. Ontologies for Detecting Conflicts

Guarino et al. describes computational ontologies as "means to formally model the structure of a system, i.e., the relevant entities and relations that emerge from its observation." [17]. It is required to conduct a mapping of statements to concepts and relationships. Inconsistencies and opposing elements can be recognized this way [18].

This shows that ontology-based methods are not easy to apply. A certain amount of preparatory work is needed, including system knowledge. The resulting advantage is that not only LNC contradictions, but also dialectic contradictions can be detected.

In this paper, however, we want to focus on LNC contradictions and lay the foundation for automatically finding contradictions in the future, without requiring system knowledge or preparatory work. Neither is expedient with ontologies.

## 4. Method for Detecting Contradictions

The findings from the Section 2 *Fundamentals* can be summarized as shown in Figure 3, while dialectical contradictions, antinomies, and subcontraries—as already explained—will not be considered:

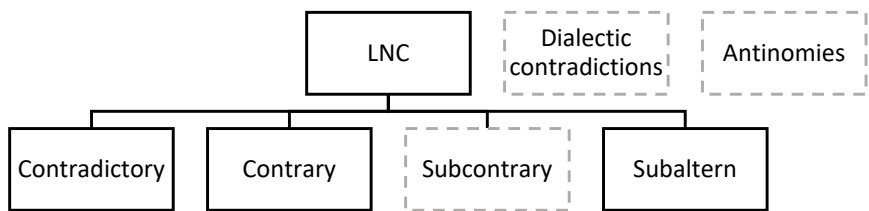

**Figure 3.** Contradictions.

In the following, we present a formal method by which contradictions can be identified and classified. To have a meaningful application in the RE context, we first must subdivide the LNC contradictions. Since many requirements are not just simple statements, we have added the principle of cause and effect to LNC, as this is not specifically represented in this theory. With this, our categories are still based on Aristotle but adapted to the Requirement context. In Section 3, they will be validated with examples from the automotive sector.

### 4.1. Nomenclature

First, we must define a nomenclature, to be able to refer to it in the following sections. Capital letters as *A*, *B*, *C*, and *D*:

- Are events that represent, for example, conditions
- Are always unequal
- Can occur simultaneously
- Do not depend on each other

Lowercase letters *x* and *y*:

- Are variables

Lowercase letters *c* and *k*:

- Are parameters
- Are unequal to each other
- Can occur in parallel: *c* can be equal to 1 and at the same time *k* equal to 2.

Operators

- $\overset{!}{=}; \overset{!}{<}; \overset{!}{>}$: must be equal, must be smaller, must be bigger
- $\Rightarrow$: implies; if... then. e.g., $A \Rightarrow x \overset{!}{=} 1$ translates to "If *A* is true, then *x* must be 1".
- $\neg$: not. e.g., The statement $\neg A$ is true if and only if *A* is false.
- $\wedge$; $\vee$: and; or. e.g., The statement $A \wedge B$ is true if *A* and *B* are both true; otherwise, it is false. Another example is: The statement $A \vee B$ is true if *A* or *B* (or both) are true; if both are false, the statement is false.

*4.2. Contradictions—Subcategories*

The suggested categories are shown in Figure 4:

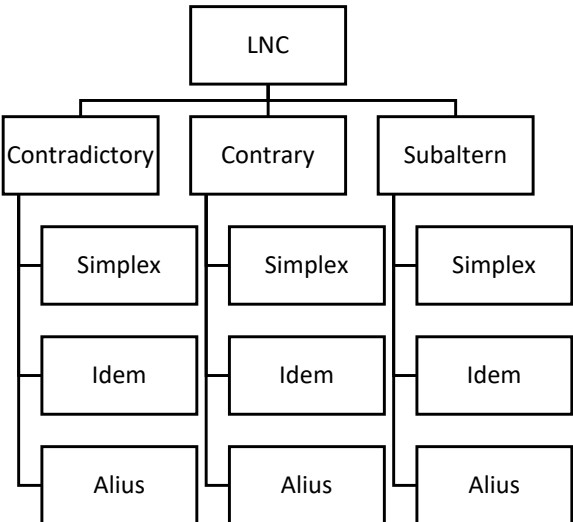

**Figure 4.** Classification of contradictions.

The terms Simplex, Idem, and Alius are described below. It should be noted that requirements can be formulated as "condition + conclusion" as well as inverted as "conclusion + condition".

Simplex (lat. = simple) refers to contradicting requirements without conditions (non-causal):

- The car must be red: $x \overset{!}{=} k$.

- The car must be blue: $x \overset{!}{=} c$.

Idem (lat. = same) refers to contradicting causal requirements with the same conditions or pairs where only one requirement has a condition:

- If the customer wishes, the car must be red: $A \Rightarrow x \overset{!}{=} k$.

- If the customer wishes, the car must be blue: $A \Rightarrow x \overset{!}{=} c$.

Alius (lat. = different) refers to contradicting causal requirements with the different conditions (causal):

- The car must be red if the customer wishes it to be: $A \Rightarrow x \overset{!}{=} k$.

- The car must be blue if the car has four doors: $B \Rightarrow x \overset{!}{=} c$.

The last example is a contradiction because the conditions of both requirements can be fulfilled at the same time (they are independent of each other) and the conclusions would then contradict each. Also, this is an example where the condition and conclusion have been inverted in order.

If one requirement is non-causal and the other is causal, the contradiction as a whole is said to be causal.

"Contradictory" refers only to the effect, not to the requirement as a whole. If the effects are contradictory but the requirements as a whole would be contrary, we still refer to them as contradictory here.

The following Table 1 lists all types of contradiction in their formalized form. The column "Multiple condition" shows examples of formalized requirements with multiple conditions. It is not an exhaustive list of all possible multiple conditions:

**Table 1.** Formalized contradictions.

| Contradictions | | Simple | Examples for Multiple Conditions |
|---|---|---|---|
| Contradictory | Simplex | $x \stackrel{!}{=} k$ <br> $x \stackrel{!}{=} \neg k$ | - |
| | Idem | $A \Rightarrow x \stackrel{!}{=} k$ <br> $A \Rightarrow x \stackrel{!}{=} \neg k$ | $A \wedge B \Rightarrow x \stackrel{!}{=} k$ <br> $A \wedge B \Rightarrow x \stackrel{!}{=} \neg k$ |
| | Alius | $A \Rightarrow x \stackrel{!}{=} k$ <br> $B \Rightarrow x \stackrel{!}{=} \neg k$ | $A \wedge B \Rightarrow x \stackrel{!}{=} k$ <br> $A \wedge C \Rightarrow x \stackrel{!}{=} \neg k$ |
| Contrary | Simplex | $x \stackrel{!}{=} c$ <br> $x \stackrel{!}{=} k$ | - |
| | Idem | $A \Rightarrow x \stackrel{!}{=} c$ <br> $A \Rightarrow x \stackrel{!}{=} k$ | $A \wedge B \Rightarrow x \stackrel{!}{=} c$ <br> $A \vee B \Rightarrow x \stackrel{!}{=} k$ |
| | Alius | $A \Rightarrow x \stackrel{!}{=} c$ <br> $B \Rightarrow x \stackrel{!}{=} k$ | $A \wedge B \Rightarrow x \stackrel{!}{=} c$ <br> $C \wedge D \Rightarrow x \stackrel{!}{=} k$ |
| Subaltern | Simplex | $x \stackrel{!}{<} c + k$ <br> $x \stackrel{!}{<} c$ | - |
| | Idem | $A \Rightarrow x \stackrel{!}{<} c + k$ <br> $A \Rightarrow x \stackrel{!}{<} c$ | $A \wedge B \Rightarrow x \stackrel{!}{<} c + k$ <br> $A \wedge B \Rightarrow x \stackrel{!}{<} c$ |
| | Alius | $A \Rightarrow x \stackrel{!}{<} c + k$ <br> $B \Rightarrow x \stackrel{!}{<} c$ | $A \wedge B \Rightarrow x \stackrel{!}{<} c + k$ <br> $C \wedge D \Rightarrow x \stackrel{!}{<} c$ |

If a condition is composed of two or-statements, it can be split into two sentences, as intermediate. This will be shown in Section 5.2.4 *Alius Contrary*. Each partial cause can then separately be considered with the effect. From $A \vee B \Rightarrow x \stackrel{!}{=} c$ follows $A \Rightarrow x \stackrel{!}{=} c$ and $B \Rightarrow x \stackrel{!}{=} c$. This facilitates the comparison of requirements that consist of compound conditions.

*4.3. Process*

The following Figure 5 shows how the types of contradiction can be recognized based on simple, but specific questions.

The first three questions refer to the effects of the requirements to be compared. The following three questions refer to the causes, if any. The questions are elaborated on below. For a contradiction to be identified, all questions must be answered as specified in the corresponding column. The check mark stands for "yes" and the cross for "no". The circle stands for questions, that do not apply in that case. Condition 1 and condition 2 are the respective conditions of the effects of requirement 1 and requirement 2. The same applies for cause 1 and cause 2.

Effect-related questions:

1. Are the variables from condition 1 and condition 2 the same or a subset of each other?

Two statements can contradict each other in the sense of LNC only if the variables, i.e. the object in question, are the same or one is a part of the other, for example, table and table leg.

2. Does one condition include the other one?

If one condition includes the other, it could be a subaltern contradiction, for example, "... between 15 m and 30 m" and "... between 20 m and 22 m". The range of the second

statement is included in the range of the first statement and therefore the former is the superaltern of the latter.

3.    Are condition 1 and condition 2 mutually exhaustive and mutually inconsistent?

This question aims at finding contradictory opposites, for example, "the car is ready"/ "the car is not ready". If one is true, the other must be false and vice versa.

Cause-related questions:

4.    Is there a condition?

If there is a condition, any form of Simplex-contradiction can be excluded.

5.    Can cause 1 occur at the same time as cause 2?

Two statements can only contradict each other, if they can theoretically occur at the same time. The statements "If it rains, ... " and "If it does not rain, ... " cannot occur at the same time and are therefore not contradicting each other. If "it rains" in the first statement were to be replaced with "it's hot outside", the two statements could theoretically contradict each other.

6.    Are cause 1 and cause 2 the same?

This question simply aims at detecting Idem-contradictions, who must have the same cause, for example, "If I am here, you are there" and "If I am here, you are here".

| Questions | | Answers | | | | | | | | |
|---|---|---|---|---|---|---|---|---|---|---|
| **Effect:** Are the variables from condition 1 and 2 the same or a subset from each other? | ✓ | ✓ | ✓ | ✓ | ✓ | ✓ | ✓ | ✓ | ✓ |
| **Effect:** Does one condition include the other one? | ✓ | ✓ | ✓ | ✗ | ✗ | ✗ | ✗ | ✗ | ✗ |
| **Effect:** Are condition 1 and condition 2 mutually exhaustive and mutually inconsistent? | ✗ | ✗ | ✗ | ✓ | ✓ | ✓ | ✗ | ✗ | ✗ |
| **Cause:** Is there a condition? | ✗ | ✓ | ✓ | ✗ | ✓ | ✓ | ✗ | ✓ | ✓ |
| **Cause:** Can cause 1 occur at the same time as cause 2? | ○ | ✓ | ✓ | ○ | ✓ | ✓ | ○ | ✓ | ✓ |
| **Cause:** Are cause 1 and cause 2 the same? | ○ | ✓ | ✗ | ○ | ✓ | ✗ | ○ | ✓ | ✗ |
| Resulting Type of contradiction | Simplex subaltern | Idem subaltern | Alius subaltern | Simplex contradictory | Idem contradictory | Alius contradictory | Simplex contrary | Idem contrary | Alius contrary |

| Legend | |
|---|---|
| ✗ | The answer to the question must be "no" |
| ✓ | The answer to the question must be "yes" |
| ○ | Answer is irrelevant |

**Figure 5.** Process overview.

## 5. Materials and Results

In this section, first, the underlying data set for the validation is explained. Afterward, the contradiction types defined above are validated using an example from the dataset, if so found. The dataset was analyzed by hand. The document was read through to find all existing contradictions. Not only contradictions, but also duplicates, repetitions, ambiguities and other conflicts were found. For a complete and automated application over a large data set, see Section *7 Conclusions*.

### 5.1. Materials

The data set consists of several interrelated requirements documents. The originator is the company IAV GmbH (Berlin, Germany), which was kind enough to make the documents available. The goal was to create a complete requirements package for the development

of E-buses, which are in use today. The document consists of about 3500 functional and non-functional requirements, from system to software level. The original language of the documents is German and was translated to English. For confidentiality issues, signal names are anonymized by using square brackets.

*5.2. Results*

Contradictory connections are counted as one contradiction. In other words: every contradictory pair is counted as a single contradiction.

From a total of 6500 objects 3500 were requirements. Besides the above mentioned other conflicts, 49 (1.35%) LNC-contradictions were found. However, it should be noted that not all contradictions were evenly distributed across all levels. 46 of the 49 contradictions were found at the software level, where they account for 2.53% of all requirements. The distribution of the different contradiction types is displayed in Table 2.

**Table 2.** Distribution.

| Simplex Subaltern | Alius Subaltern | Alius Contradictory | Alius Contrary |
|:---:|:---:|:---:|:---:|
| 4 | 3 | 2 | 40 |

These figures must be viewed with caution, as the analysis was done manually, and it is likely that further inconsistencies were overlooked.

We didn't find any contradictions for the following species: Simplex and Idem Contradictories, Idem Contraries, and Idem Subalterns. This will be reflected in Section 6 *Discussion*. In the following, we explain the method using one example each from the requirements documents.

5.2.1. Simplex Subaltern

The two selected requirements are:

1.　The safe state must be reached within 1000 ms.
2.　The safe state must be reached within 800 ms.

The building blocks are shown in Figure 6:

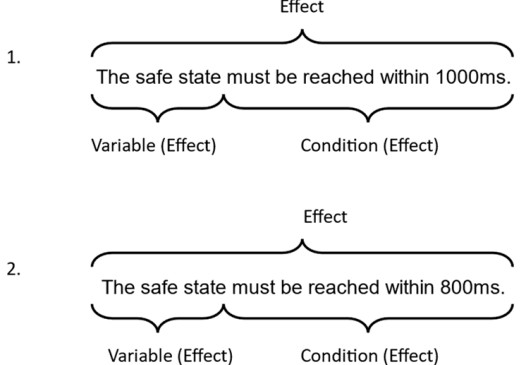

**Figure 6.** Building blocks for a Simplex Subaltern contradiction, consisting of two requirements.

Its formalized form is:

$$x \overset{!}{<} k \tag{1}$$

$$x \overset{!}{<} c \tag{2}$$

$$\textit{while } c < k \tag{3}$$

where "safe state" is $x$, "1000 ms" is $k$ and "800 ms" is $c$.

The questions presented in our methodology can then be answered as shown in Figure 7:

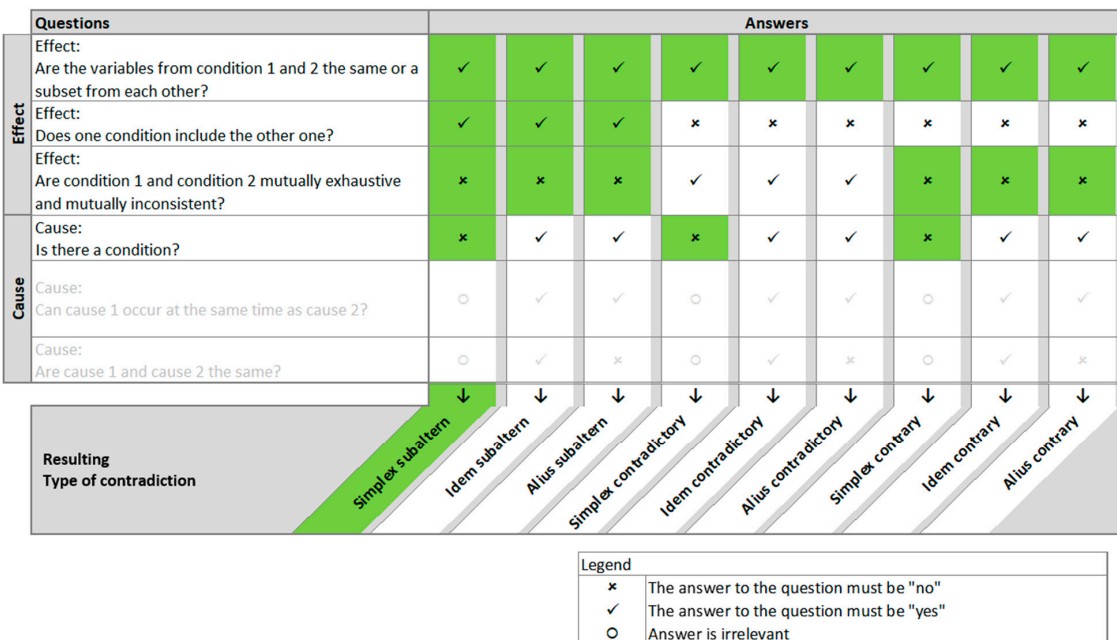

**Figure 7.** Process for Simplex Subaltern.

The questions were answered as given in the column for Simplex Subaltern contradictions. The last two cause-questions did not need to be answered, because the Simplex-contradictions do not have causes.

5.2.2. Alius Subaltern

The two selected requirements are:

1.  If the actual heater stage CbnHeatg_[ ... ] > 0, the requested pump power CbnHeatg_SpOfCooltPmp must be limited by the parameter CbnHeatg_TrigForDutyCyc Of[ ... ].
2.  If BattChrgnMngt_MsgVld[ ... ] = false, the requested pump power CbnHeatg_SpOf CooltPmp must be limited to 20%.

The parameter CbnHeatg_TrigForDutyCycOf[...] is initialized elsewhere with 80%. Therefore, we have a similar case as above, only this time there are conditions. The building blocks are shown in Figure 8:

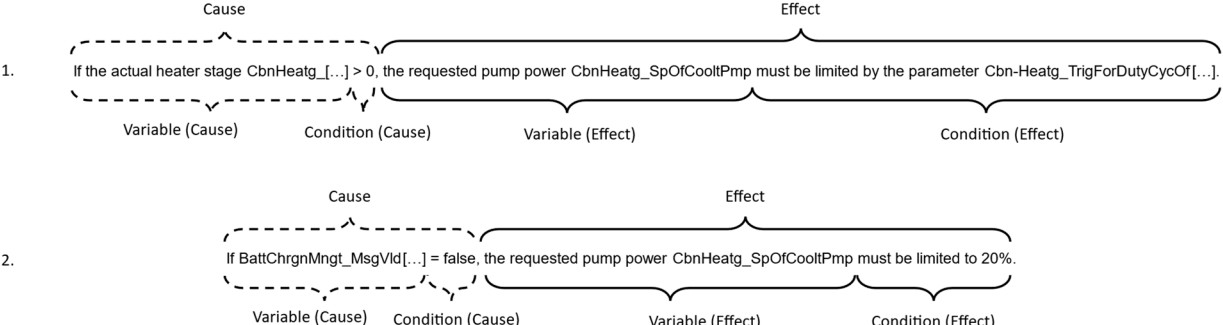

**Figure 8.** Building blocks for an Alius Subaltern contradiction, consisting of two requirements.

And results in:

$$A \Rightarrow x \overset{!}{<} k \tag{4}$$

$$B \Rightarrow x \overset{!}{<} c \tag{5}$$

$$while \; c < k \tag{6}$$

where ''If the actual heater stage CbnHeatg_[ . . . ] > 0'' is $A$, "If BattChrgnMngt_Msg Vld[ . . . ] = false" is $B$, "CbnHeatg_SpOfCooltPmp" is $x$, "Cbn-Heatg_TrigForDutyCyc Of[ . . . ]" is $k$ and "20%" is $c$.

The questions presented in our methodology can then be answered as shown in Figure 9:

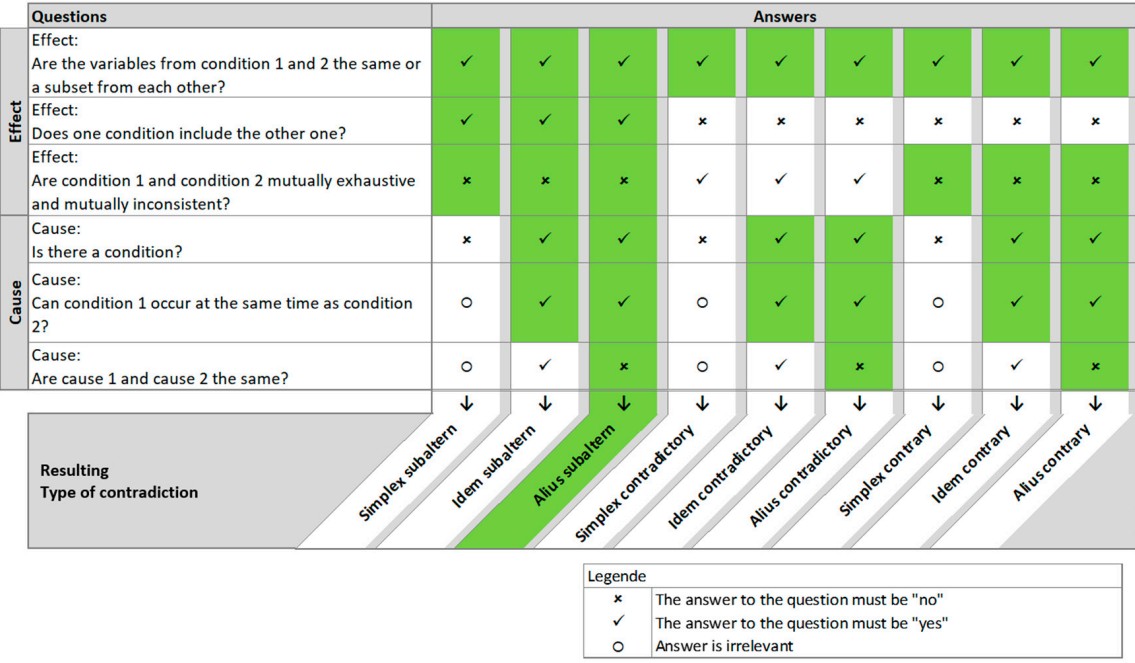

**Figure 9.** Process for Alius Subaltern.

### 5.2.3. Alius Contradictory

It gets more complicated when getting to the following contradictories:

1.  Suitable potential equalization is required for all conductive covers or housings of all HV components.
2.  If additional external conductive sheaths or covers are fitted over covers or enclosures consisting of solid insulating materials, equipotential bonding is not required for these.

By considering the context, it becomes clear, that the demonstrative "these" in the second sentence is a variable $y$. It refers to "covers or housings consisting of insulating materials" and not to "covers or housings" or "solid insulating materials". However, the variable $x$ of the first sentence is "covers or housings", which means that $y \in x$.

Therefore, the building blocks are as shown in Figure 10:

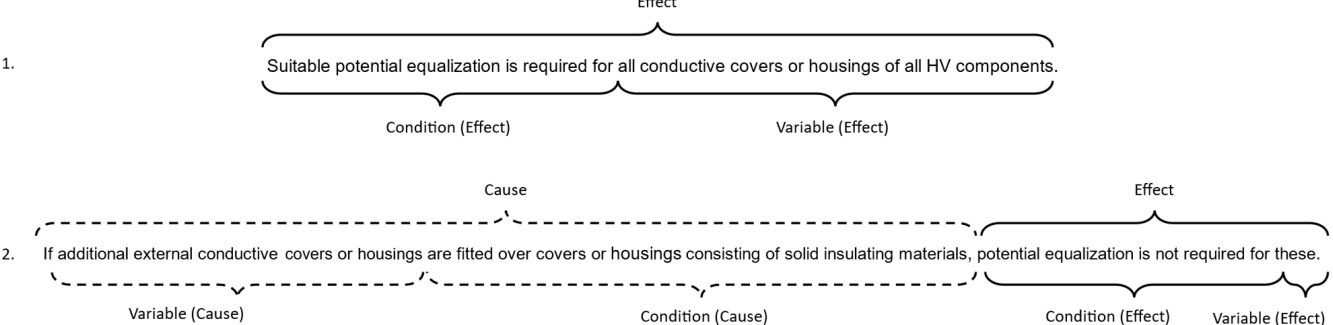

**Figure 10.** Building blocks for an Alius Contradictory contradiction, consisting of two requirements.

The formalized form is:

$$x \overset{!}{=} k \tag{7}$$

$$A \vee B \Rightarrow y \overset{!}{\neq} k \tag{8}$$

$$while\ y \in x \tag{9}$$

where "conductive covers or housings of all HV components" is $x$, "potential equalization" is $k$, "additional external conductive covers or housings are fitted over covers" is $A$ and "housings consisting of solid insulating materials" is $B$. "these" is $y$ and is actually a subset of $x$. It denotes "conductive covers or housings of all HV components with additional external conductive covers or housings fitted over covers or housings consisting of solid insulating materials".

After the formalized form has been determined, filling in the table works as usual, as shown in Figure 11:

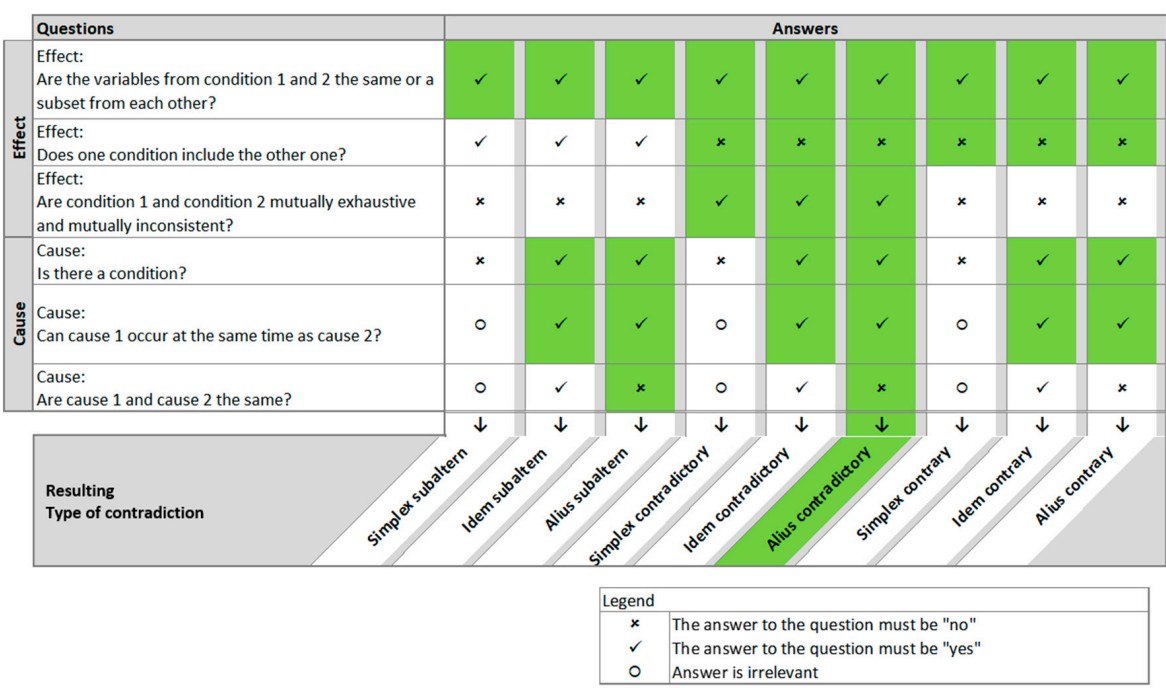

**Figure 11.** Process for Alius Contradictory.

### 5.2.4. Alius Contrary

The two selected requirements are:

1. If the value of the signal ComVehFrnt_ChrgnCur[ … ] exceeds the value of 0 (A), the signal Chrgn[ … ] must be set to TRUE.
2. If the parameter ChrgnCurChk_SubVal[ … ] is set to TRUE, the signal Chrgn[ … ] corresponds to the parameterizable value ChrgnCurChk_SubValChrgn[ … ], otherwise, the signal is forwarded unchanged.

The second condition of the second requirement should be transferred to a separate requirement to apply this method. The second requirement thus splits and can be checked separately against other requirements for contradictions. Accordingly, our customized requirements look like this, while we will be using 2.1 in the further analysis:

2.1 If the parameter ChrgnCurChk_SubVal[ … ] is set to TRUE, the signal Chrgn[ … ] corresponds to the parameterizable value ChrgnCurChk_SubValChrgn[ … ].

2.2 If the parameter ChrgnCurChk_SubVal[ … ] is *not* set to TRUE, the signal Chrgn[ … ] is forwarded unchanged.

Then, the building blocks are as shown in Figure 12:

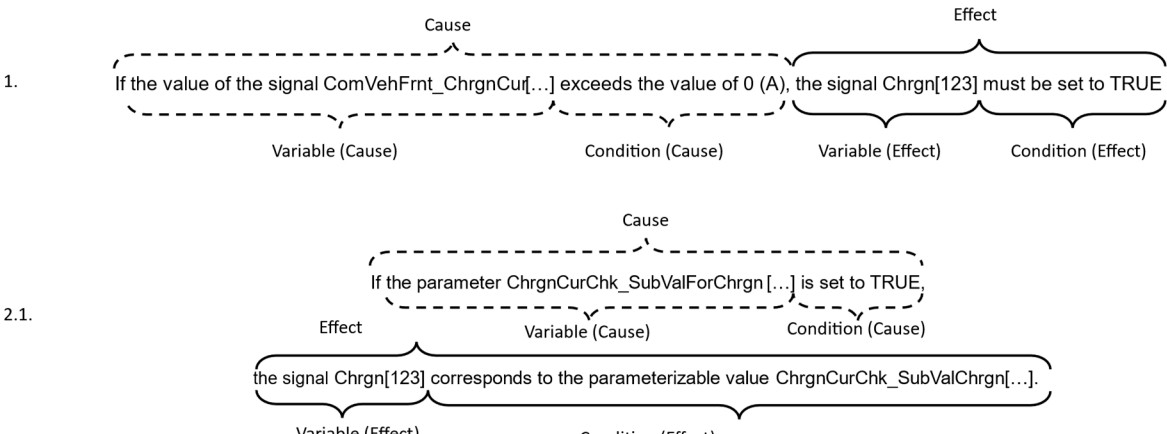

**Figure 12.** Building blocks for an Alius Contrary contradiction, consisting of two requirements.

The formalized form results in:

$$A \Rightarrow x \stackrel{!}{=} c \tag{10}$$

$$B \Rightarrow x \stackrel{!}{=} k \tag{11}$$

where "the value of the signal ComVehFrnt_ChrgnCur[ ... ] exceeds the value of 0 (A)" is *A*, "the parameter ChrgnCurChk_SubValForChrgn[ ... ] is set to TRUE" is *B*, "Chrgn[123]" is *x* and "TRUE" is *c* and "ChrgnCurChk_SubValChrgn[ ... ]" is *k*.

The questions presented in our methodology can then be answered as shown in Figure 13:

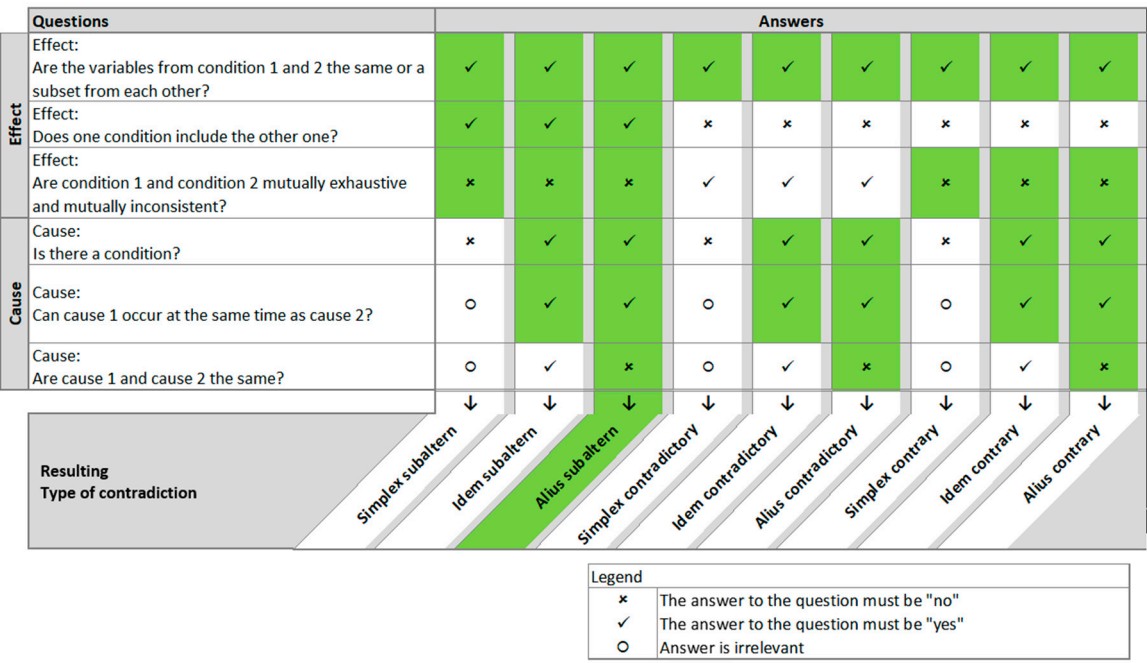

**Figure 13.** Process for Alius Contrary.

## 6. Discussion

It is important to note, that we didn't find any Idem-contradictions and only one Simplex-contradiction. Idem-contradictions are so conspicuous that the requirements engineer would probably notice them immediately since he would have to formulate exactly the same cause twice with the same variables but different effects. The reason for the absence of Simplex-contradictions is, that the examined system is so complex

that simple statements without conditions would simply not be sufficient to describe the system precisely.

Besides mentioned reasons, internal validity mistakes could play a role in not finding certain contradiction types. In the project documents are about 3500 requirements with $\sum_{k=0}^{n-1} k = 6,123,250$ theoretical combinations. We therefore cannot rule out the possibility, that we missed Idem- or Simplex-contradictions.

If the requirements are not formulated according to the guidelines, borderline cases can certainly occur in which contradictions cannot be clearly assigned or even identified. This is because language is often ambiguous and human interpretation is often needed. When it comes to complex formulations, even common sense can reach its limits.

## 7. Conclusions

Especially in the early development phase, ambiguities are very common in Requirements documents due to the use of natural language. In this paper, we examined contradictive requirements, which we defined using formal logic. In contrast to other papers, we did not classify contradictions according to our data set or our code, but according to a generally accepted, well-tested systematic model. Then, we created a classification tailored to RE, in which conditions and effects now take a prominent role. Finally, we proposed a way to identify our contradictions using clear questions.

We have analyzed about 6500 objects, approximately 3500 of which were requirements. In total, we were able to identify many different conflicts, 49 of which were LNC-related contradictions that could be identified using our method. The majority of the detected contradictions were of the Alius Contraries. Furthermore, most of the contradictions were found at the deeper system levels, namely those of the software requirements. This corresponds to our expectations, since requirements on the higher levels are written less concretely and describe the general functionality of the product. As a result, there is often no risk of contradictions in the first place.

With our method, contradictions can be found in a semi-automated way: The classification into cause and effect, as well as variable and condition, are fully automated, for example by using Fischbach's parser [19]. This way our method can be applied automatically up to the step "Building Blocks". In the then following formalization the building blocks must be replaced by symbols and formulas. However, this step is not automated. To the best of our knowledge, there are currently no methods available which allow for this. Therefore, further research is required, as mentioned in Section 6 *Discussion*. Once the formalization is done, answering the questions in Figure 5 presents a simple—yet manual—task. This was shown with examples in Section 5.2 *Results*.

When applying our method, a requirements reviewer does not have to be familiar with requirements in general or with the topic of the document anymore, to recognize contradictions, as our method provides a simple recipe for detecting LNC-related contradictions.

Future work could entail automation, quality analysis and non-LNC-contradictions. Requirements documents can become very extensive due to the necessary level of detail [20]. Therefore, an automated determination of contradictions would be useful. The formalization of contradictions proposed in this paper provides strong implications for automation, by serving as the basis for a fully automated contradiction-detection method. The queries that would have to be made in such a code are already mathematically formulated here.

We can also derive implications for an automated quality analysis. The classification into different types of contradictions is an important step to quantify the quality of a requirements document. The logical next step would be to assess the criticality of the contradiction. Based on this, a meaningful key performance indicator could be determined. This would require analyzing a large number of inconsistencies, to assess the impact on the product, as well as any different resolution methods per type of contradiction. The greater the impact and the more difficult the solution, the more critical the contradiction.

As we saw in Section 2 *Fundamentals*, there are other types of contradictions besides LNC-contradictions, that have not been discussed in this paper: dialectic contradictions and

antinomies. In our opinion, dialectic contradictions cannot be detected by applying simple rules, instead, they require context and language comprehension. It might be possible to achieve results with a sufficiently large and clean data set and by using machine learning algorithms. Regarding antinomies, it should first be checked whether they occur at all in requirements documents. A solution to these contradictions is similar to the solution of dialectical contradictions.

**Author Contributions:** Conceptualization, A.E.G., D.G. and T.-A.F.; methodology, A.E.G.; validation, A.E.G.; formal analysis, A.E.G.; investigation, A.E.G.; resources, A.E.G. and D.G.; data curation, A.E.G.; writing—original draft preparation, A.E.G.; writing—review and editing, D.G. and T.-A.F.; supervision, D.G. All authors have read and agreed to the published version of the manuscript.

**Funding:** We acknowledge support by the German Research foundation and the Open Access Publication Fund of TU Berlin.

**Data Availability Statement:** Not applicable.

**Conflicts of Interest:** The authors declare no conflict of interest.

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
