# Peer review of "Fundamental Research on Detecting Contradictions in Requirements: Taxonomy and Semi-Automated Approach"

_applsci, doi:10.3390/app12157628_

Round 1

Reviewer 1 Report

The article describes a taxonomy for contradictions in requirement engineering documents and applies it to real life industrial requirement documents.

Section 1:

Line 59: Explain the abbreviation DOORS. What is a DOORS document?

Line 63: Could be elaborated a bit more. What is meant by boilerplate and placeholder values here? If it is not relevant, consider removing it.

Section 2:

Line 194-213: Nomenclature is hard to read. Consider adding some whitespace to seperate the different symbols.

Line 208: In the rest of the paper you exlusively use a "must be smaller" operator. Consider using "must be smaller" here as well instead of or in addition to "must be bigger".

Line 209: seems wrong or poorly worded.  A => B is true when A and B is true, but also if A is false. The current description sounds identical to A ^ B.

Line 229: Clarify that this includes pairs of requirements where only one requirement has conditions.

Table 1: Capital letters V and W are not explained in nomenclature.

Table 1: Alius Contradictory (Simple). Choice of letter "C" instead of "B" seems confusing as it might be considered deliberate. Should be consistent with the rest of the table.

Table 1: The "multiple conditions" are not really explained in text. It is unclear if these multiple conditions are only examples or if this is an exhaustive list of all possible multiple conditions for the contradiction type.

Line 245-249 seems to describe another case than section 3.2.5. In the former the effect is the same (x=c), in the latter they are not. Also in the former there are 2 independent causes, while in the latter the causes are clearly dependent (A vs NOT A).

Line 262: Origin of "condition 1" and "condition 2" are not clearly explained in text. It should be clarified that these are the respective conditions of the effects of a requirement 1 and 2.

Section 3:

Line 289: Colaborate a bit on how the analysis by hand was done. Did you try to find one example for each case of contradiction or did you try to find all contradictions? If the latter, can this be quantified and included in the paper (e.g. how many counts of each contradiction). If it is the former, what was the method, searching for keywords/terms in the document?

Line 351: It is not immediately clear what A and B stand for in the formalized form. The formalized form could use some additional explanation.

Line 374: By your own nomenclature two capital letters A and B are events that are independent of each other. This is clearly not the case here since B = NOT A.

Section 4:

Line 387-396: The paragraph sounds like automated reasoning is easily done using NLP parsers, while this is a very hard problem. Just looking at some of your examples it is hard to see how NLP could reliable extract a formalized form that is suitable for automated reasoning. While it is not the aim of your article to provide a solution to this, I think it could be more clearly stated as a limitation and challenge for automatization.

Line 408: The paper only considers combinations of two requirements. 3000 requirements therefore would results in 3000*2999 possible pairings, not 3000!.

Tables and Figures:

Tables 2-6: Not sure if this is a mistake or technical limitation, but the legend caption uses german word "Legende".

Tables 2-6 as well as some of the figures contain text in bitmap graphics and are therefore hard to read at some zoom levels. Consider using vector graphics instead.

General feedback:

Title: The title promises a "semi-automated approach", but it seems that there was no automatization involved. Consider clarifying the title or how the approach used automatization.

You introduce requirements as having cause and effect both of which can have conditions. Then you introduce LNC and contradictions. Both are well written and easy to understand individually. However you only really combine both in the concrete examples, which makes it hard to follow. E.g. it is not immediately obvious how to conditions in effects are represented in LNC logic.

To judge the validity and usefulness of the approach it would be helpful to get a sense of how many contradictions are estimated to be present in industrial requirement documents, how many of these fall in the scope of LNC and how many of these could potentially be detected by your approach. If there is no data on this in the literature or in your own research, this could be mentioned as a challenge, limitation, and/or future work.

Author Response

Section 1:

Line 59: Explain the abbreviation DOORS. What is a DOORS document?

DOORS is a Requirements-Management-Tool for requirements in tabular form. Still, I deleted the example, as it was causing confusion.

Line 63: Could be elaborated a bit more. What is meant by boilerplate and placeholder values here? If it is not relevant, consider removing it.

I tried to elaborate for a better understanding.

Section 2:

Line 194-213: Nomenclature is hard to read. Consider adding some whitespace to seperate the different symbols.

I changed the formatting

Line 208: In the rest of the paper you exlusively use a "must be smaller" operator. Consider using "must be smaller" here as well instead of or in addition to "must be bigger".

I added “must be bigger”

Line 209: seems wrong or poorly worded.  A => B is true when A and B is true, but also if A is false. The current description sounds identical to A ^ B.

It was indeed poorly worded. I changed it accordingly.

Line 229: Clarify that this includes pairs of requirements where only one requirement has conditions.

I added it by rephrasing the sentence.

Table 1: Capital letters V and W are not explained in nomenclature.

I replaced it with B and C

Table 1: Alius Contradictory (Simple). Choice of letter "C" instead of "B" seems confusing as it might be considered deliberate. Should be consistent with the rest of the table.

I replaced it with B

Table 1: The "multiple conditions" are not really explained in text. It is unclear if these multiple conditions are only examples or if this is an exhaustive list of all possible multiple conditions for the contradiction type.

It is indeed not an exhaustive list of multiple conditions. I changed it accordingly.

Line 245-249 seems to describe another case than section 3.2.5. In the former the effect is the same (x=c), in the latter they are not. Also in the former there are 2 independent causes, while in the latter the causes are clearly dependent (A vs NOT A).

Maybe my explanation was confusing. I tried to elaborate more as follows: In section 3.2.5 I am comparing requirement 1 with requirement 2.1. In this case, the causes are independent. In table 1 the causes are also independent. The formalized forms are different in 1 and 2.1. I changed the order of c and k in the formalized result, so that it matches the order in table 1. If this does not answer your note, could you elaborate on your question?

Line 262: Origin of "condition 1" and "condition 2" are not clearly explained in text. It should be clarified that these are the respective conditions of the effects of a requirement 1 and 2.

I added it: “Condition 1 and condition 2 are the respective conditions of the effects of requirement 1 and requirement 2. The same applies for cause 1 and cause 2.”

Section 3:

Line 289: Colaborate a bit on how the analysis by hand was done. Did you try to find one example for each case of contradiction or did you try to find all contradictions? If the latter, can this be quantified and included in the paper (e.g. how many counts of each contradiction). If it is the former, what was the method, searching for keywords/terms in the document?

Added in chapter 3:

The dataset was analyzed by hand:. The document was read through to find all existing contradictions. Not only contradictions, but also duplicates, repetitions, ambiguities and other conflicts were found

[…]

From a total of 6500 objects 3500 (I had to adjust the number) were requirements. Besides the above mentioned other conflicts, 49 (1.35%) LNC-contradictions were found. However, it should be noted that not all contradictions were evenly distributed across all levels. 46 of the 49 contradictions were found at the software level, where they account for 2.53% of all requirements. The distribution of the different contradiction types is displayed in Table 3.

Table 3: Distribution

Simplex Subaltern

Alius Subaltern

Alius Contradictory

Alius Contrary

4

3

2

40

These figures must be viewed with caution, as the analysis was done manually and it is likely that inconsistencies were overlooked.

Line 351: It is not immediately clear what A and B stand for in the formalized form. The formalized form could use some additional explanation.

I added explanations for all formalized forms

Line 374: By your own nomenclature two capital letters A and B are events that are independent of each other. This is clearly not the case here since B = NOT A.

With the additional explanations (see answer above) of what A and B are, hopefully it becomes more clear, that A and B are independent of each other: “the value of the signal ComVehFrnt_ChrgnCur[…] exceeds the value of 0 (A)” is A, “the parameter ChrgnCurChk_SubValForChrgn[…] is set to TRUE” is B

Section 4:

Line 387-396: The paragraph sounds like automated reasoning is easily done using NLP parsers, while this is a very hard problem. Just looking at some of your examples it is hard to see how NLP could reliable extract a formalized form that is suitable for automated reasoning. While it is not the aim of your article to provide a solution to this, I think it could be more clearly stated as a limitation and challenge for automatization.

I have edited the paragraph accordingly: However, this step is not automated. To the best of our knowledge, no methods are available which make this possible. Therefore further research is required, as mentioned in section 4.2 Future work. Today, manual work is therefore still necessary.

Line 408: The paper only considers combinations of two requirements. 3000 requirements therefore would results in 3000*2999 possible pairings, not 3000!.

Yes, I indeed made a mistake. With the adjusted number of analyzed objects should be 3500*(3500-1)/2 = 6,123,250

Tables and Figures:

Tables 2-6: Not sure if this is a mistake or technical limitation, but the legend caption uses german word "Legende".

I corrected it

Tables 2-6 as well as some of the figures contain text in bitmap graphics and are therefore hard to read at some zoom levels. Consider using vector graphics instead.

I replaced the figures in section 3 witch vector graphics. Unfortunately I couldn’t create vector graphics for the tables, since Excel seems not to support that.

General feedback:

Title: The title promises a "semi-automated approach", but it seems that there was no automatization involved. Consider clarifying the title or how the approach used automatization.

I added a conclusion, where I try to justify to title. If you think it it still not justified, I will adjust the title.

You introduce requirements as having cause and effect both of which can have conditions. Then you introduce LNC and contradictions. Both are well written and easy to understand individually. However you only really combine both in the concrete examples, which makes it hard to follow. E.g. it is not immediately obvious how to conditions in effects are represented in LNC logic.

In section 2, line 192-197, I tried to explain raw LNC-contradictions are not sufficient. To be applied in RE context, we need a structure which gets into cause+effect, as requirements often have a cause. Furthermore I added that the principle cause+effect “is not specifically represented in this theory”, to make clear, why we need to understand both principles.

To judge the validity and usefulness of the approach it would be helpful to get a sense of how many contradictions are estimated to be present in industrial requirement documents, how many of these fall in the scope of LNC and how many of these could potentially be detected by your approach. If there is no data on this in the literature or in your own research, this could be mentioned as a challenge, limitation, and/or future work

I added a conclusion, where I try to get into this.

Thank you very much for your review and feedback!

Reviewer 2 Report

Thank you for submitting your work to Applied Sciences. This paper presents a method that provides a taxonomy for automated contradiction detection research and automated quality analysis of application documents. Although the paper presents a highly interesting topic, it is recommended to extend the results obtained. It is also important to create a conclusion section of the paper in addition to the "Discussion" section, in order to concisely highlight the results obtained. Finally, in the detection method, a code was made, it would be essential to add a flow chart with the corresponding explanation.

Author Response

Thank you for submitting your work to Applied Sciences. This paper presents a method that provides a taxonomy for automated contradiction detection research and automated quality analysis of application documents. Although the paper presents a highly interesting topic, it is recommended to extend the results obtained.

I added additional information about the dataset and on the overall results (section 3-3.2):

  • How the dataset was analyzed
  • How big the dataset was
  • How many conflicts and how many contradictions were found in total
  • How the contradictions are distributed

Furthermore I tried to explain in more detail, how the results were obtained using our method (section 3.2.1-3.2.5)

It is also important to create a conclusion section of the paper in addition to the "Discussion" section, in order to concisely highlight the results obtained.

I added a section “conclusion”.

Finally, in the detection method, a code was made, it would be essential to add a flow chart with the corresponding explanation.

I didn’t make a code yet. This paper lays the foundation, for a later automation. I apologize for the confusion.

Thank you for your review and feedback!

Reviewer 3 Report

The topic is interesting. However, the manuscript should be re-organized. English writing should be polished. The results should be more clearly explained.

Author Response

The topic is interesting. However, the manuscript should be re-organized. English writing should be polished. The results should be more clearly explained.

I edited the whole paper. Especially:

  • I added additional information about the dataset and on the overall results (section 3-3.2):
    • How the dataset was analyzed
    • How big the dataset was
    • How many conflicts and how many contradictions were found in total
    • How the contradictions are distributed
  • Furthermore, I tried to explain in more detail, how the results were obtained using our method (section 3.2.1-3.2.5)
  • I added a section “conclusion”.
  • English writing was polished

If further adjustments are needed, I would kindly ask you if you could be more specific about what needs to be improved.

Thank you for your review and time!

Round 2

Reviewer 3 Report

The authors have addressed all the comments. Therefore, it can be accepted in the current form.

Author Response

The authors have addressed all the comments. Therefore, it can be accepted in the current form.

Thank you!